# OpenReview forum: "AttnGCG: Enhancing Jailbreaking Attacks on LLMs with Attention Manipulation"
_ICLR.cc/2025/Conference — ICLR 2025 Conference Withdrawn Submission_

### Official Review · Reviewer_jGoS · 2024-10-27

**Soundness:** 3
**Presentation:** 3
**Contribution:** 3
**Rating:** 5
**Confidence:** 4

**Summary:**

The paper proposes an enhancement to the GCG algorithm by integrating attention scores to improve jailbreak attacks on LLMs. By incorporating attention mechanisms, the authors aim to identify and manipulate tokens that have a higher impact on the model's output, thereby enhancing the effectiveness of the attack.

Strengths:
1. Relevant Topic: The paper addresses the important area of robustness and adversarial attacks on LLMs, which is crucial for understanding and improving AI safety.
2. Logical Extension: Integrating attention scores into the GCG algorithm is a reasonable approach, as attention mechanisms are central to how LLMs process information.

Weaknesses:
1. Limited Novelty: The primary contribution is the addition of attention scores to the existing GCG algorithm. This is an incremental improvement rather than a fundamentally new approach, since there are already paper that have suggested incorporating attentions cores to the loss function including https://arxiv.org/pdf/2001.06325, https://aclanthology.org/2023.findings-emnlp.716.pdf and more. The paper under review does not sufficiently differentiate its approach from these existing methods or provide substantial advancements over them. To be considered a significant contribution, the paper could:
Introduce a novel methodology for manipulating attention that leads to substantially improved attack success rates.
Provide theoretical insights into the vulnerabilities of attention mechanisms in LLMs.
Offer a comprehensive comparative analysis with existing attention-based attack methods to highlight unique benefits.

2. Marginal Improvements: The empirical results show only slight enhancements in attack effectiveness compared to the original GCG algorithm. The performance gains may not be substantial enough to warrant publication. The empirical results indicate only slight enhancements over the original GCG algorithm. For instance, the attack success rate on average improved from 74.1% to 78.0%, a marginal increase that may not be statistically significant or practically impactful. In adversarial attack research, improvements are generally expected to be substantial (e.g., exceeding a 10% increase in success rate) to warrant publication. Additionally, the integration of attention scores adds computational complexity to the algorithm. This includes the overhead of computing attention weights and manipulating them during the attack process, which can increase the time and resources required. The paper should provide a detailed analysis of this trade-off, justifying whether the modest performance gains outweigh the additional complexity introduced.


3. Unrealistic Experimental Settings: The experiments are conducted with a temperature setting of zero, leading to deterministic outputs from the LLM. In practical applications, models often operate with higher temperatures (e.g., between 0.7 and 1.0) to generate more diverse and creative responses. Using a temperature of zero may limit the generalizability of the results to real-world scenarios where outputs are stochastic.

It is recommended to conduct additional experiments with non-zero temperature settings to evaluate the robustness and effectiveness of the proposed attack under more realistic conditions. Analyzing whether the attention manipulation strategy remains effective when the model's outputs are less predictable would strengthen the paper. If there is theoretical justification for the method's performance at higher temperatures, including this analysis would be beneficial.

**Strengths:**

* Relevant Topic: The paper addresses the important area of robustness and adversarial attacks on LLMs, which is crucial for understanding and improving AI safety.
* Logical Extension: Integrating attention scores into the GCG algorithm is a reasonable approach, as attention mechanisms are central to how LLMs process information.

**Weaknesses:**

* Limited Novelty: The primary contribution is the addition of attention scores to the existing GCG algorithm. This is an incremental improvement rather than a fundamentally new approach, since there are already paper that have suggested incorporating attentions cores to the loss function including https://arxiv.org/pdf/2001.06325, https://aclanthology.org/2023.findings-emnlp.716.pdf and more. The paper does not sufficiently demonstrate how this modification offers a significant advancement over existing methods.

* Marginal Improvements: The empirical results show only slight enhancements in attack effectiveness compared to the original GCG algorithm. The performance gains may not be substantial enough to warrant publication, especially considering the added complexity.

* Unrealistic Experimental Settings: The experiments are conducted with a temperature setting of zero, resulting in deterministic outputs from the LLM. In practical applications, models often operate with higher temperatures to allow for more diverse and creative responses. Using a temperature of zero may limit the generalizability of the results to real-world scenarios.

**Questions:**

* Assessment of Performance Improvements: The empirical results show only slight enhancements over the original GCG algorithm. Can you provide additional analysis to demonstrate the practical significance of these performance gains? How do you justify the added complexity of integrating attention scores given the marginal improvements observed?

* Experimental Settings and Real-World Applicability: Since your experiments use a temperature setting of zero, which leads to deterministic outputs, how do you address the applicability of your findings to real-world scenarios where higher temperatures are common? Have you conducted experiments with non-zero temperature settings, and if so, could you share those results to illustrate the robustness of your method under more practical conditions?

---

### Official Review · Reviewer_TBiK · 2024-11-02

**Soundness:** 2
**Presentation:** 2
**Contribution:** 2
**Rating:** 3
**Confidence:** 3

**Summary:**

This paper highlights the susceptibility of Large Language Models (LLMs) to adversarial attacks despite existing safety measures. Traditional strategies like Greedy Coordinate Gradient (GCG) often fail to effectively optimize the model's internal focus. The authors propose that AttnGCG manipulates attention scores to strengthen attack success, redirecting the model's focus from safety prompts to adversarial inputs. The paper conducts extensive empirical studies and justifies the effectiveness of the proposed method.

**Strengths:**

- The AttnGCG forces LLM to pay more attention to the adversarial suffix, inducing more harmful contents from the model, which is novel and effective.
- Extensive experiments have been conducted to provide a good insight into the components of the proposed method.
- The paper is generally well-written, with clear illustrations and tables.

**Weaknesses:**

- The paper focuses on the development and success of the adversarial attack technique without sufficiently exploring how such attacks can be mitigated or defended against. This leaves a gap in the practical applicability of the research, as understanding potential defenses is crucial to improving model robustness in real-world scenarios.
- As discussed in the limitation part, the AttenGCG failed to jailbreak the latest LLM like GPT-4o. The low transferability of GCG and AttenGCG shows the low risk of the reveal flaw.
- The AttnGCG still relies on exclusive optimization on white-box LLM. Excessive resource requirements could also act as a defense method. It would be more attractive to manipulate the attention matrix directly to jailbreak the LLM.

**Questions:**

1. Discuss the effectiveness of the defense method for the AttenGCG, such as the perplexity defense [1] adopted by AutoDAN.
2. Discuss the possibility of manipulating the attention matrix directly to jailbreak the LLM.

[1] Jain, Neel, et al. "Baseline defenses for adversarial attacks against aligned language models." arXiv preprint arXiv:2309.00614 (2023).

---

### Official Review · Reviewer_gDCt · 2024-11-03

**Soundness:** 3
**Presentation:** 3
**Contribution:** 2
**Rating:** 5
**Confidence:** 5

**Summary:**

This paper introduces a new strategy called AttnGCG, which builds on the existing Greedy Coordinate Gradient (GCG) method. It focuses on improving jailbreaking attacks on large language models (LLMs) by manipulating attention scores to optimize adversarial prompts. The study finds that increasing attention to adversarial suffixes can enhance attack success rates (ASR) and demonstrates that this approach outperforms traditional GCG in both direct and transfer attacks.

**Strengths:**

1. The study utilizes the attention mechanism of LLMs to guide the optimization of jailbreak prompts, offering a new perspective on how adversarial inputs can bypass safety measures.

2. The authors perform evaluations in both white-box and transfer attack settings, showcasing the robustness and transferability of AttnGCG.

3. Attention heatmap visualizations enhance the interpretability of the attack process, providing insights into why specific jailbreak attempts succeed.

4. The presentation is clear and well-structured, making it easy to follow and understand the main contributions of this work.

**Weaknesses:**

1. The improvements of AttnGCG over GCG are relatively limited: The experimental results indicate that the performance gains of AttnGCG over GCG are under 10% in most cases.

2. Limitations of baseline comparisons: The experiments only compare AttnGCG with GCG. To strengthen the paper's credibility, it would be beneficial for the authors to include comparisons with other existing jailbreak attack methods, such as AutoDAN [1], COLD-Attack [2], AdvPrompter [3], and AmpleGCG [4]. This would help position AttnGCG's relative advantages and uniqueness within the current research landscape.

3. Lack of evaluation under safety system prompts: The experiments focus solely on the attack efficacy without considering scenarios where safety system prompts are in use. To validate the real-world applicability of AttnGCG, testing its performance with active safety system prompts, especially in comparison with methods from [5,6,7], would help determine if it maintains or improves upon the advantages over GCG under enhanced security measures.

4. Applicability to the latest models: Although the paper mentions that AttnGCG is less effective on the latest models (e.g., Gemini-1.5-Pro-latest and GPT-4o), there is no in-depth analysis or discussion provided. More research and data are needed to explore potential improvements or strategies to address these limitations, ensuring better applicability and robustness in future versions.

[1] Liu X, Xu N, Chen M, et al. Autodan: Generating stealthy jailbreak prompts on aligned large language models[J]. arXiv preprint arXiv:2310.04451, 2023.

[2] Guo X, Yu F, Zhang H, et al. Cold-attack: Jailbreaking llms with stealthiness and controllability[J]. arXiv preprint arXiv:2402.08679, 2024.

[3] Paulus A, Zharmagambetov A, Guo C, et al. Advprompter: Fast adaptive adversarial prompting for llms[J]. arXiv preprint arXiv:2404.16873, 2024.

[4] Liao Z, Sun H. Amplegcg: Learning a universal and transferable generative model of adversarial suffixes for jailbreaking both open and closed llms[J]. arXiv preprint arXiv:2404.07921, 2024.

[5] Xie Y, Yi J, Shao J, et al. Defending chatgpt against jailbreak attack via self-reminders[J]. Nature Machine Intelligence, 2023, 5(12): 1486-1496.

[6] Zhang Z, Yang J, Ke P, et al. Defending large language models against jailbreaking attacks through goal prioritization[J]. arXiv preprint arXiv:2311.09096, 2023.

[7] Zhou Y, Han Y, Zhuang H, et al. Defending jailbreak prompts via in-context adversarial game[J]. arXiv preprint arXiv:2402.13148, 2024.

**Questions:**

The same as the weaknesses.

---

### Official Review · Reviewer_PCnt · 2024-11-08

**Soundness:** 2
**Presentation:** 3
**Contribution:** 2
**Rating:** 3
**Confidence:** 4

**Summary:**

This paper proposes enhancements to GCG jailbreak attacks. The central observation is that by increasing the model's attention to adversarial suffix, the effectiveness of the attack can be significantly improved.

**Strengths:**

* The paper addresses a critical and relevant problem in the field of large language models (LLMs), emphasizing the necessity of studying jailbreak attacks.
* The authors evaluate their approach on both open-source and closed-source models, providing a broader perspective on the attack's effectiveness.
* The paper leverages the attention mechanism inherent in transformer architectures to design more effective attacks, showcasing a novel approach to enhancing jailbreak techniques.

**Weaknesses:**

1. The choice to target optimization-based jailbreak attacks raises questions, as this type often does not achieve higher success rates compared to alternatives (e.g., LLM-aided jailbreaking) and is more susceptible to defenses (e.g., simple perplexity analysis can defend against this type of attacks).
2. The introduction references multiple figures, some of which appear on subsequent pages, disrupting the flow of reading and comprehension.
3. The evaluation primarily compares against the GCG attack, which is relatively outdated. To substantiate the effectiveness of their approach, the authors should include comparisons with more recent state-of-the-art methods.
4. The paper lacks a thorough analysis of existing literature that also utilizes the attention mechanism in transformer models for attack purposes. This oversight diminishes the contextual understanding of their contributions.
5. The decision to use only 100 harmful questions from AdvBench, rather than the complete dataset of 520 prompts, raises questions about the representativeness and robustness of their findings.
6. While the paper notes that outputs generated by GCG do not necessarily constitute valid jailbreak attacks—even with a high probability of harmful tokens (lines 50-53)—it fails to propose a solution to this issue. Furthermore, reliance on GPT-4 for output evaluation is not a novel contribution, as it is commonly used in other studies. Additionally, Table 1 indicates that jailbreak suffixes generated by AttnGCG also exhibit significant false positives in terms of jailbreak classification.

**Questions:**

See above

---

### Note · Authors · 2024-11-18

I have read and agree with the venue's withdrawal policy on behalf of myself and my co-authors.